# TAMING GRADIENT OVERSMOOTHING AND EXPANSION IN GRAPH NEURAL NETWORKS

## ABSTRACT

Oversmoothing has been claimed as a primary bottleneck for multi-layered graph neural networks (GNNs). Multiple analyses have examined how and why oversmoothing occurs. However, none of the prior work addressed how optimization is performed under the oversmoothing regime. In this work, we show the presence of *gradient oversmoothing* preventing optimization during training. We further analyze that GNNs with residual connections, a well-known solution to help gradient flow in deep architecture, introduce *gradient expansion*, a phenomenon of the gradient explosion in diverse directions. Therefore, adding residual connections cannot be a solution for making a GNN deep. Our analysis reveals that constraining the Lipschitz bound of each layer can neutralize the gradient expansion. To this end, we provide a simple yet effective normalization method to prevent the gradient expansion. An empirical study shows that the residual GNNs with hundreds of layers can be efficiently trained with the proposed normalization without compromising performance. Additional studies show that the empirical observations corroborate our theoretical analysis.

## 1 INTRODUCTION

Increasing depths of layers is recognized as a key reason for the success of many deep neural networks. Despite this widely believed claim, the graph neural network (GNN) does not necessarily benefit from deeper architectures. In fact, increasing the number of layers in GNNs often leads to problems such as oversmoothing, where node representations from different classes become indistinguishable, resulting in a degradation of performance. Although residual connections contribute to the success of deep architectures in many types of neural networks, they are generally ineffective at resolving the oversmoothing in GNNs.

To provide a deeper understanding of the oversmoothing, several studies have addressed the mechanism behind the oversmoothing. Li et al. (2018) explain that the graph convolutional network (GCN) is a special form of Laplacian smoothing. Chamberlain et al. (2021) show that GNN can be interpreted as a discretization of the heat diffusion process, implying the diffusion of the node representation to reach equilibrium. Cai & Wang (2020) and Oono & Suzuki (2020) theoretically show that oversmoothing occurs exponentially when piling the layers in GCN under certain conditions. Wu et al. (2023a) show Graph Attention Networks (GATs), a generalization of GCN, also lose their expressive power exponentially with the number of layers. However, none of the previous work on oversmoothing investigates the gradient flow of the deep GNNs.

In this work, we investigate gradient flows under the oversmoothing regime. Understanding the exact characteristics of gradients can help mitigate the oversmoothing problem. To do this, we first analyze the gradients with the graph convolutional network (GCN). The analysis shows the presence of *gradient oversmoothing*, which implies a gradient of node features are similar to each other. We then investigate the effect of the residual connections in GNNs, which is known to make the optimization work with many variants of the neural networks. Interestingly, we show that the opposite effect, named *gradient expansion*, outbreaks as a side effect of the residual connection. When the gradient expansion occurs, the gradients of node features explode in diverse directions. We find that the gradient expansion is tightly connected with the upper bound of the Lipschitz constants of GNN layers. To relax the gradient expansion, we propose a simple weight normalization that controls the Lipschitz upper bound.

Empirical studies on node classification tasks corroborate our theoretical analysis, showing that gradient smoothing and gradient expansion happen with deep GNNs and deep residual GNNs, respectively. We also show that the similarity between gradients better represents the test performance than the similarity between node representations, a commonly used oversmoothing measure. Finally, we present how the deep GNNs perform with the Lipschitz constraints. With the Lipschitz constraints, we can successfully train the GNNs with residual connections without sacrificing the test accuracy on node classification tasks.

## 2 RELATED WORK

Recently, many studies have investigated the reason for performance degradation in deep GNNs. Oversmoothing, the phenomenon where the representations of neighboring nodes become similar as the number of GNN layers increases, has been identified as one of the main reasons. Several studies have theoretically demonstrated that oversmoothing occurs due to the structural characteristics of message-passing GNNs. Li et al. (2018) reveal that GCN is a special form of Laplacian smoothing and points out that repeated applications lead to convergence to the same value. Oono & Suzuki (2020) demonstrate that, as the depth of the GCN architecture increases, the distance between representations and invariant spaces decays exponentially, leading to oversmoothing. Keriven (2022) show that, in linear GNNs whose aggregation matrix takes the form of a stochastic matrix, the output converges to the average value of the training labels as the depth approaches infinity. Wu et al. (2023b) analyze the mixing and denoising effects of GNN layers on graphs sampled from the contextual stochastic block model, providing insight into why oversmoothing can occur even before depth approach infinity. Wu et al. (2023a) prove that oversmoothing occurs exponentially even in the graph attention network architectures with non-linear activations.

Based on these analyses, many studies try to mitigate oversmoothing to stack multiple layers without performance degradation. PairNorm (Zhao & Akoglu, 2020) keeps the total pairwise representation distances constant by normalization. DropEdge (Rong et al., 2020) randomly drops edges from the graph to slow down oversmoothing. Energetic graph neural networks (Zhou et al., 2021) utilize the Dirichlet energy term, which measures oversmoothing in the loss function to regularize oversmoothing. Ordered GNN (Song et al., 2023) preserves aggregated representation within specific hops separately. Gradient Gating (Rusch et al., 2023b) proposes a mechanism to stop learning in a node-wise fashion before local oversmoothing occurs. Park et al. (2024) propose to reverse the aggregation process. Some studies propose to change the dynamics of GNN, based on the findings of Chamberlain et al. (2021), which shows that classical GNN resembles diffusion-like dynamics. GRAND++ (Thorpe et al., 2022) proposes to add a source term to prevent oversmoothing. PDE-GCN (Eliasof et al., 2021) proposes wave-like GNN architecture. Graph-coupled oscillator networks (Rusch et al., 2022) propose a network based on non-linear oscillator structures.

Some studies propose to improve the optimization process for multi-GNN layer stacking. Li et al. (2018) adopt residual connection to improve the training process. Zhang et al. (2022) also point out model degradation as a main reason for the performance degradation of deep GNNs and proposes an adaptive initial residual to improve optimization. However, no studies explain the exact reason for model degradation and residual connection's effectiveness.

## 3 PRELIMINARY

### 3.1 GRAPH CONVOLUTIONAL NETWORKS

We begin by considering undirected graphs $G = (\mathcal{V}, \mathcal{E}, \mathbf{X})$, where $\mathcal{V}$ represents the set of $N \in \mathbb{N}$ *nodes*, $\mathcal{E} \subseteq \mathcal{V} \times \mathcal{V}$ forms the *edge* set, a symmetric relation, and $\mathbf{X} \in \mathbb{R}^{N \times d}$ is a collection of $d$-dimensional feature matrix for each node. We use $\mathbf{X}_i \in \mathbb{R}^d$ to represent the feature vector of node $i$. The connectivity between nodes can also be represented through an *adjacency matrix* $\mathbf{A} \in \{0, 1\}^{N \times N}$. We focus on a graph convolutional network (GCN) (Kipf & Welling, 2017) and its variants. With an initial input feature matrix $\mathbf{X}^{(0)} := \mathbf{X}$, at each layer $0 \leq \ell < L$, the *representation of nodes* $\mathbf{X}^{(\ell)}$ is updated as follows:

$$\mathbf{X}^{(\ell+1)} := \sigma(\hat{\mathbf{A}}\mathbf{X}^{(\ell)}\mathbf{W}^{(\ell)}), \tag{1}$$

where $\hat{\mathbf{A}} = \tilde{\mathbf{D}}^{-\frac{1}{2}}\tilde{\mathbf{A}}\tilde{\mathbf{D}}^{-\frac{1}{2}}$ is an normalized adjacency matrix with $\tilde{\mathbf{A}} = \mathbf{A} + \mathbf{I}$, diagonal matrix $\tilde{D}_{ii} = \sum_j \tilde{A}_{ij}$, $\sigma$ is non-linear activation function, and $\mathbf{W}^{(\ell)}$ a layer specific learnable weight matrix. Linear Graph Neural networks (LGN) represent the GCN without an activation function.

There are several variations of GCN through different definitions of the normalized adjacency matrix. For example, Graph Attention Networks (GATs) use a learnable adjacency matrix through an attention layer (Veličković et al., 2018). Regardless of the precise formulation of the normalized matrix, most variations share similar characteristics; the node representations are updated by aggregating the neighborhood representations.

The residual connections can be incorporated into the GNN to help the gradient flow with many layers. The update rule of the node representation with the residual connection is as follows:

$$\mathbf{X}^{(\ell+1)} := \sigma(\hat{\mathbf{A}}\mathbf{X}^{(\ell)}\mathbf{W}^{(\ell)}) + \mathbf{X}^{(\ell)}. \tag{2}$$

### 3.2 SIMILARITY MEASURE

We follow the definition of node similarity measure of node feature matrix $\mathbf{X}$ defined in Wu et al. (2023a) as

$$\mu(\mathbf{X}) := \|\mathbf{X} - \mathbf{1}\gamma_X\|_F, \text{ where } \gamma_X = \frac{\mathbf{1}^\top \mathbf{X}}{N}, \tag{3}$$

and $\|\cdot\|_F$ indicates Frobenius norm. Considering $\mathbf{B} \in \mathbb{R}^{(N-1)\times(N-1)}$, the orthogonal projection into the space perpendicular to span $\mathbf{1}$, the definition of $\mu$ satisfies $\mu(\mathbf{X}) = \|\mathbf{B}\mathbf{X}\|_F$. This definition of node similarity satisfies the following two axioms (Rusch et al., 2023a):

1. $\exists \mathbf{c} \in \mathbb{R}^d$ such that $\mathbf{X}_i = \mathbf{c}$ for all node $i$ if and only if $\mu(\mathbf{X}) = 0$, for $\mathbf{X} \in \mathbb{R}^{N \times d}$;
2. $\mu(\mathbf{X} + \mathbf{Y}) \leq \mu(\mathbf{X}) + \mu(\mathbf{Y})$, for all $\mathbf{X}, \mathbf{Y} \in \mathbb{R}^{N \times d}$.

*Representation oversmoothing* with respect to $\mu$ is then characterized as the layer-wise convergence of the node similarity measure $\mu$ to zero, i.e.,

$$\lim_{\ell \to \infty} \mu\left(\mathbf{X}^{(\ell)}\right) = 0.$$

The node similarity measure can further be used to compute the similarity between the gradients of node features. Let $\frac{\partial \mathcal{L}}{\partial \mathbf{X}^{(\ell)}} \in \mathbb{R}^{N \times d}$ be the partial derivatives w.r.t some loss function $\mathcal{L}$ given input feature matrix and output labels. With the same measure $\mu$, the gradient similarity at layer $\ell$ can be computed by $\mu\left(\frac{\partial \mathcal{L}}{\partial \mathbf{X}^{(\ell)}}\right)$. *Gradient smoothing* can happen when the gradient similarity converges to zero. We further characterize the *gradient expansion*, which describes the cases where the gradient similarity of the first layer diverges, i.e.,

$$\lim_{L \to \infty, \ell \to 1} \mu\left(\frac{\partial \mathcal{L}}{\partial \mathbf{X}^{(L-\ell)}}\right) = \infty.$$

Since $\|\mathbf{X}\|_F^2 = \mu(\mathbf{X})^2 + \|\mathbf{X} - \mathbf{B}\mathbf{X}\|_F^2$, gradient expansion implies a gradient explosion. We note that, however, gradient expansion and gradient explosion are not the same concept. The gradient expansion occurs when there is a *gradient explosion in diverse directions* and differs from the gradient explosion where we only measure the magnitude of the gradients.

## 4 GRADIENT OVERSMOOTHING AND EXPANSION

It is widely recognized that increasing the depth of graph neural networks (GNNs) typically results in performance degradation. While oversmoothing has been identified as a key cause, questions remain about why this issue cannot be resolved through optimization. One possible explanation is that optimization is inherently tricky in GNNs, such as gradient vanishing and exploding, commonly observed in the earlier stages of neural networks. However, it is also observed that the performance of deep GNNs has limitations with the more advanced techniques known to improve optimization, such as the residual connections (He et al., 2016).

To answer these questions, we thoroughly analyze the gradient of the linear GCNs with and without residual connections first. The analysis shows the presence of gradient oversmoothing that is similar

to the representation oversmoothing but a fundamental reason why the training is incomplete with deep GCNs. Our analysis further identifies that gradient expansion is a more common problem with the residual connections, explaining the failure of the residual connections in deep GCNs. We then extend our analysis of the GNNs with a non-linear activation through empirical experiments to show that gradient oversmoothing and expansion can also be observed in practice. We conduct our experiment also on GATs, showing that gradient oversmoothing and expansion is not the only problem in GCN, although our theoretical analyses are limited to GCN.

## 4.1 THEORETICAL ANALYSIS

To understand how the optimization is performed withGCNs, we first analyze the gradients of the GCN during training. First, let us show the exact gradients of the parameter matrix at each layer during training.

**Lemma 1.** *(Blakely et al., 2021) Let* LGN *be* $L$-*layered linear GCN without an activation function, and* $\mathcal{L}_{\mathsf{LGN}}(\mathbf{W}, \mathbf{X}, \mathbf{y})$ *be a loss function of the linear GCN with a set of parameters* $\mathbf{W} = (\mathbf{W}^{(0)}, \cdots, \mathbf{W}^{(L-1)})$ *and input feature matrix* $\mathbf{X}$ *and output label* $\mathbf{y}$. *The gradient of the parameter at layer* $\ell$, *i.e.,* $\mathbf{W}^{(\ell)}$, *with respect to the loss function is*

$$\frac{\partial \mathcal{L}_{\mathsf{LGN}}}{\partial \mathbf{W}^{(\ell)}} = (\hat{\mathbf{A}}\mathbf{X}^{(\ell)})^{\top} \frac{\partial \mathcal{L}_{\mathsf{LGN}}}{\partial \mathbf{X}^{(\ell+1)}} \ ,$$

*where*

$$\frac{\partial \mathcal{L}_{\mathsf{LGN}}}{\partial \mathbf{X}^{(\ell)}} = (\hat{\mathbf{A}}^{\top})^{L-\ell} \frac{\partial \mathcal{L}_{\mathsf{LGN}}}{\partial \mathbf{X}^{(L)}} \prod_{i=1}^{L-\ell} (\mathbf{W}^{(L-i)})^{\top} \ .$$

Lemma 1 shows that the formulation of gradient $\frac{\partial \mathcal{L}_{\mathsf{LGN}}}{\partial \mathbf{W}^{(\ell)}}$ resembles the forward propagation steps of the LGN. Specifically, $\frac{\partial \mathcal{L}_{\mathsf{LGN}}}{\partial \mathbf{X}^{(\ell)}}$ can be interpreted as the update equation of $(L - \ell)$-layered LGN with $\frac{\partial \mathcal{L}_{\mathsf{LGN}}}{\partial \mathbf{X}^{(L)}}$ as an input feature matrix, and $\hat{\mathbf{A}}^{\top}$ and $\mathbf{W}^{(\ell)\top}$ as a normalized adjacency and weight matrix at layer $\ell$, respectively. Consequently, following the general analysis of the oversmoothing in node representations, we can show the oversmoothing in gradients of the GNN. The following theorem characterized the upper bound on the similarity between the gradient of different nodes in a deeper layer.

**Theorem 1** (Gradient oversmoothing in LGN). *With the same assumptions used in Lemma 1 and a additional assumption that the graph $G$ is connected and non-bipartite, there exists $0 < q < 1$ and constant $C_q > 0$ such that*

$$\mu \left( \frac{\partial \mathcal{L}_{\mathsf{LGN}}}{\partial \mathbf{X}^{(\ell)}} \right) \leq C_q \left( q \|\mathbf{W}^{(*)}\|_2 \right)^{(L-\ell)}, \quad 0 \leq \ell < L \ , \tag{4}$$

*where $\|\mathbf{W}^{(*)}\|_2$ is the maximum spectral norm, i.e., $* = \arg\max_i \|\mathbf{W}^{(i)}\|_2$ for $1 \leq i \leq L - \ell$.*

Theorem 1 implies that the similarity measure of the gradient decreases exponentially while approaching to the first layer if $q\|\mathbf{W}^{(*)}\|_2 < 1$. In other words, as the number of depths increases, the gradient information required to update the layers closer to the input becomes oversmoothed regardless of the input feature. When $q\|\mathbf{W}^{(*)}\|_2 > 1$, a gradient expansion occurs. As a result, optimization becomes extremely difficult in GNNs with deeper layers due to the structural characteristics.

Although Theorem 1 shows the challenges in GCN training, similar challenges are also prevalent in any neural network architectures. The skip connection or residual connection is the off-the-shelf solution to improve the gradient flows in deep architectures (He et al., 2016). The residual connections are also suggested in previous studies to improve the performance of GNNs with multiple layers (Li et al., 2018). Although they successfully stacked 56 layers, this is relatively shallow compared to ResNet, which stacks more than 152 layers. To better understand the gradient flow with the residual connections in GCN, we first state the exact gradients of the parameter matrix at each layer.

**Lemma 2.** *Let* resLGN *be a linear graph neural network with residual connections at each layer, and* $\mathcal{L}_{\mathsf{resLGN}}(\mathbf{W}, \mathbf{X}, \mathbf{y})$ *be a loss function of* resLGN *with a set of parameters* $\mathbf{W} =$

$(\mathbf{W}^{(0)}, \cdots, \mathbf{W}^{(L-1)})$ *and input feature matrix* $\mathbf{X}$ *and output label* $\mathbf{y}$*. The gradient of the parameter at layer* $\ell$*, i.e.,* $\mathbf{W}^{(\ell)}$*, with respect to the loss function is*

$$\frac{\partial \mathcal{L}_{\mathsf{resLGN}}}{\partial \mathbf{W}^{(\ell)}} = \left(\hat{\mathbf{A}}\mathbf{X}^{(\ell)}\right)^{\top} \frac{\partial \mathcal{L}_{\mathsf{resLGN}}}{\partial \mathbf{X}^{(\ell+1)}} \,,$$

*where*

$$\frac{\partial \mathcal{L}_{\mathsf{resLGN}}}{\partial \mathbf{X}^{(\ell)}} = \frac{\partial \mathcal{L}_{\mathsf{resLGN}}}{\partial \mathbf{X}^{(L)}}$$

$$+ \sum_{p=1}^{L-\ell} \left( \underbrace{\sum_{\{i_1 : i_1 \geq \ell\}} \sum_{\{i_2 : i_2 > i_1\}} \cdots \sum_{\{i_p : i_{p-1} < i_p < L\}}}_{\textit{all possible length-p paths in back-propagation}} \left(\hat{\mathbf{A}}^{\top}\right)^{p} \frac{\partial \mathcal{L}_{\mathsf{resLGN}}}{\partial \mathbf{X}^{(L)}} \prod_{k=0}^{p-1} \left(\mathbf{W}^{(i_{p-k})}\right)^{\top} \right) . \tag{5}$$

Compared to Lemma 1, the gradient of resLGN is expressed as the summation of $L - \ell + 1$ terms, each of which considers all possible length $p$ back-propagation paths. Each path resembles the forward propagation rule of the LGN similar to Lemma 1. As a result, the gradients are unlikely to be oversmoothed, as terms with shorter path lengths that are not oversmoothed still remain. However, the issue of gradient explosion still exists as the number of layers $L$ increases.

The following theorem states the upper bound of the gradient similarity w.r.t. node representation at layer $\ell$.

**Theorem 2** (Gradient expansion in resLGN). *With the same assumptions used in Lemma 2, there exists* $0 < q < 1$ *and constant* $C_q > 0$*, such that*

$$\mu\left(\frac{\partial \mathcal{L}_{\mathsf{resGNN}}(\mathbf{W})}{\partial \mathbf{X}^{(\ell)}}\right) \leq \mu\left(\frac{\partial \mathcal{L}_{\mathsf{resGNN}}(\mathbf{W})}{\partial \mathbf{X}^{(L)}}\right) + C_q \left(\left(1 + q\|\mathbf{W}^{(*)}\|_2\right)^{L-\ell} - 1\right), \quad 0 \leq \ell < L \,, \tag{6}$$

*where* $* = \arg\max_i \|\mathbf{W}^{(i)}\|_2$ *for* $1 \leq i \leq L - \ell$*.*

Theorem 2 implies that the gradients can *expand* with the residual connections when $q\|\mathbf{W}^{(*)}\|_2 > 0$. However, unlike in LGNs, the upper bound on gradient similarity cannot converge to zero unless $q\|\mathbf{W}^{(*)}\|_2$ is zero. Therefore, with the residual connection, the gradient oversmoothing is less of a concern than the gradient expansion.

**Lipschitz upper bound constraint.** The Lipschitz upper bound of a GCN layer is $\|\mathbf{W}\|_2$ since renormalization trick makes $\|\tilde{\mathbf{A}}\|_2 = 1$ (Kipf & Welling, 2017). Therefore, Theorem 2 implies that we can stabilize the training procedure and mitigate the gradient explosion by controlling the Lipschitz constant of the GCN layer. The exact $\|\mathbf{W}\|_2$ is the largest singular value of the parameter matrix, which is often difficult to compute. In this work, we use a simple alternative to control the upper bound on the Lipschitz constant by normalizing the weight through its Frobenius norm, i.e., $\mathbf{W} \leftarrow c\mathbf{W}/\|\mathbf{W}\|_F$, where $c$ is a hyperparameter controlling the upper bound of the Lipschitz constant as shown in Park et al. (2024). We note that since the largest singular value of the right stochastic matrix corresponds to one, we can apply the same approach in GAT. In our experiments, we normalize the weight matrices at each training iteration. The efficiency of the normalization is shown in Section 6.

## 4.2 EMPIRICAL STUDIES

We conduct an empirical study to show that the theoretical findings of linear GNNs can also be observed in non-linear GNNs. For the experiments, we use GCN and GAT and their residual counterparts, resGCN, and resGAT, respectively, as baseline models. To train the models, we use two homophilic datasets, Cora and CiteSeer, and one heterophilic dataset, Chameleon. For each dataset, we train 64- and 128-layer GCN, GAT, resGCN, and resGAT. We consider three activation functions: ReLU, LeakyReLU with 0.8 negative slope values, and GeLU. We measure the gradient similarity when the test performance is measured through validation. All experiments are repeated five times, and their average values are reported. The band indicates min-max values.

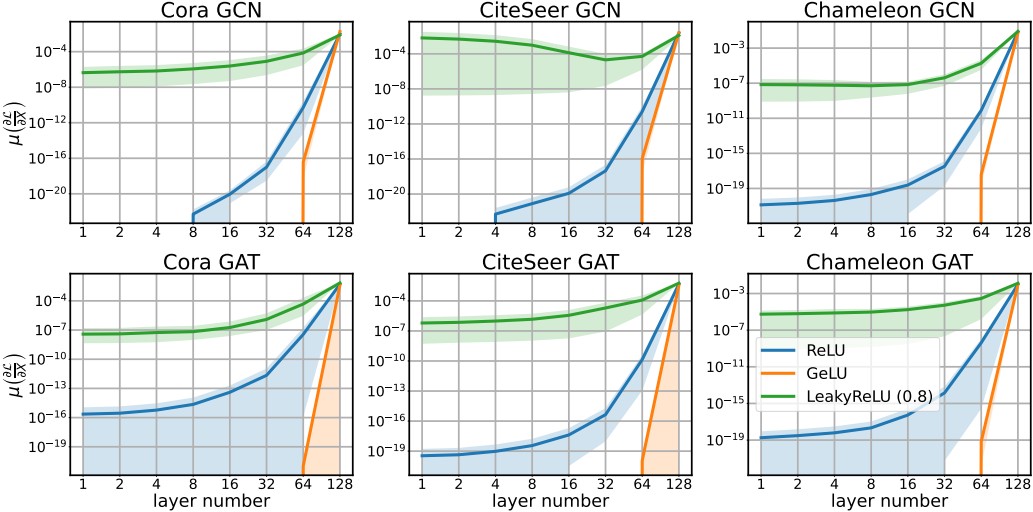

Figure 1: Gradient similarity measure $\mu\left(\frac{\partial \mathcal{L}_{\text{GNN}}(\mathbf{W})}{\partial \mathbf{X}^{(\ell)}}\right)$ over different layers and activation functions of 128-layer GCN and GAT in three datasets: Cora, CiteSeer, and Chameleon.

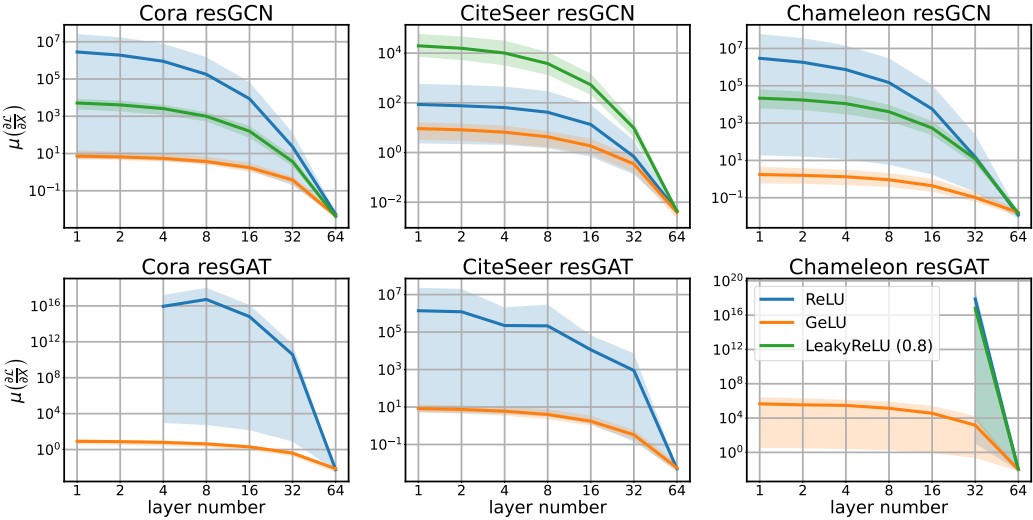

Figure 2: Gradient similarity measure $\mu\left(\frac{\partial \mathcal{L}_{\text{resGNN}}(\mathbf{W})}{\partial \mathbf{X}^{(\ell)}}\right)$ over different layers of 64-layer GCN and GAT with residual connections in three datasets: Cora, CiteSeer, and Chameleon. The similarity measures with "NaN" value are not indicated in the plot.

**GNNs without residual connections** Figure 1 illustrates that the gradients of node representations are getting exponentially close to each other over the back-propagation paths for both 128-layer GCN and GAT, independent of the activation functions and datasets. In other words, all nodes are likely to have the same gradient signals near the input layers. Among three activation functions, GELU leads to the most severe oversmoothing in gradients, followed by ReLU and LeakyReLU. The experimental results align with the one with node representations but in the other direction (Wu et al., 2023a). Additionally, we report the results with 64-layer GCN and GAT in Appendix C.

💡 In deep GNNs, the node representations are getting similar near the final layers *(representation oversmoothing)*, and the node gradients are getting similar near the input layers due to the oversmoothing effects *(gradient oversmoothing)*.

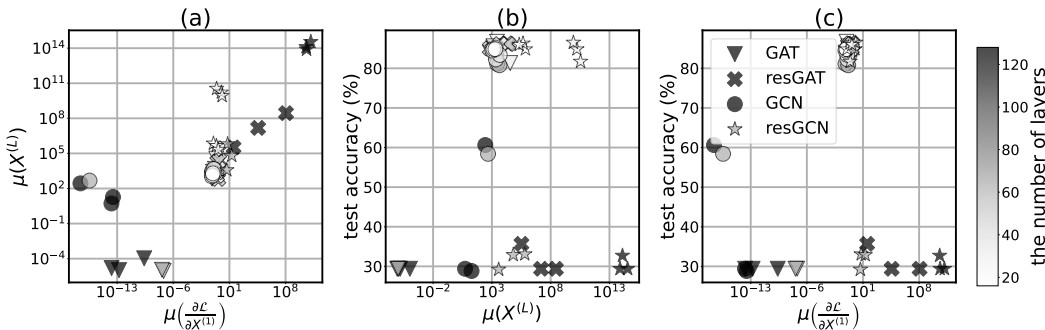

Figure 3: Scatter plots between (a) representation similarity vs gradient similarity (b) test accuracy and representation similarity, (c) test accuracy and gradient similarity on the Cora dataset.

**GNNs with residual connections**  We visualize the gradient similarity of 64-layer resGCN and resGAT in Figure 2 with varying activation functions. The results show that, with the residual connection, the gradients expand regardless of the activations, datasets, and models, while the expansion rate differs depending on the choice of activation, model, and dataset. Note that gradient expansion happens when the magnitude of the gradient is large, and the gradient diverges in directions. However, among all experiments, we could not find any gradient smoothing, showing that gradient expansion is a more severe problem with residual connections. In conclusion, these results corroborate the theoretical analysis in Theorem 2. Additional results with 128-layer resGCN and resGAT are provided in Appendix C.

> 💡 With residual connections, the gradients of GNN expand as the depth of the layer increases *(gradient expansion)*. The gradient oversmoothing is barely observed with residual connections.

To explore the training behavior of GNN models, we trained 4-, 16-, and 64-layer GCN, GAT, resGCN, and resGAT models on three datasets: Cora, Citeseer, and Chameleon, with a fixed learning rate of $0.001$ for $1,000$ epochs. Visualizations of the training accuracy and training loss are provided in Appendix D. We observe that both GCN and GAT, with and without residual connections, suffer from more severe underfitting as the number of layers increases. We infer that this phenomenon is related to the presence of gradient oversmoothing and gradient expansion.

## 5 ANALYSIS ON GRADIENT SIMILARITY

We have seen gradients exhibit oversmoothing or expansion based on the model architecture. In this section, we further analyze the relationship between representation similarity and gradient similarity and the changes in the gradient similarity over the training.

### 5.1 WHICH SIMILARITY MEASURE BETTER REPRESENTS TEST PERFORMANCE?

To obtain a deeper insight into the representation and gradient similarities and their implications on model performances, we train GCN, GAT, resGCN, and resGAT with varying numbers of layers on the Cora and Chameleon datasets and observe empirical similarities and model performances. The node similarity is measured from the representation of the last layer, and the gradient similarity is measured from the first layer of the model. All similarities are measured when the test performances are measured through validation. We plot the empirical relation between 1) representation similarity, 2) gradient similarity, and 3) accuracy on the test set in Figure 3 through scatter plots.

Figure 3(a) shows the relationship between the representation similarity and the gradient similarity. We observe that, in general, there are positive correlations between these two similarities, but they cannot fully explain each other in all regions. For example, when the number of layers is relatively low, the gradient similarity ranges around one while the node similarity ranges between $10^2$ and

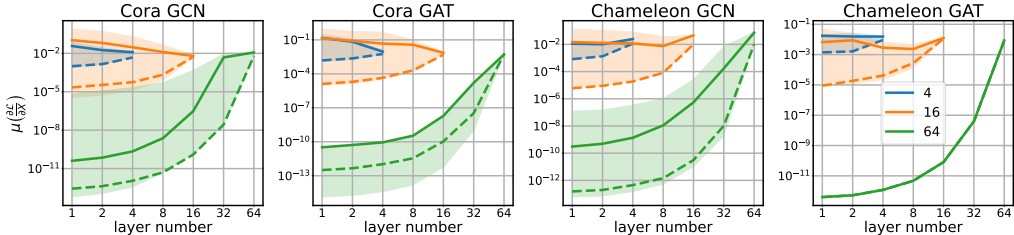

Figure 4: Gradient similarity measure $\mu \left( \frac{\partial \mathcal{L}_{\text{GNN}}(\mathbf{W})}{\partial \mathbf{X}^{(\ell)}} \right)$ over training with 4-, 16-, 64- layered GCN and GAT. We report the similarity measures over two datasets: Cora and Chameleon. The dashed and solid lines represent the similarity measured at the start and end of the training, respectively. The shaded area represents the maximum and minimum similarities over training.

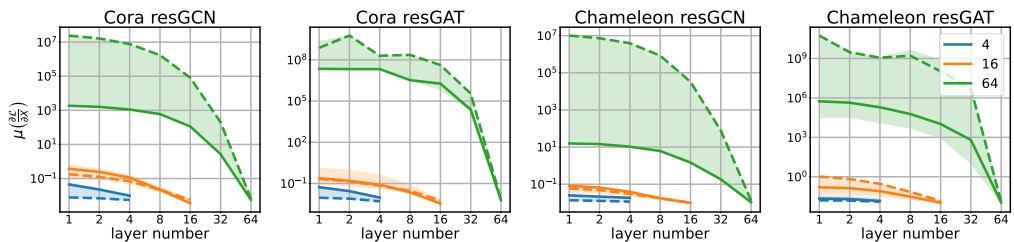

Figure 5: Gradient similarity measure $\mu \left( \frac{\partial \mathcal{L}_{\text{GNN}}(\mathbf{W})}{\partial \mathbf{X}^{(\ell)}} \right)$ over training with 4-, 16-, 64- layered resGCN and resGAT. The dashed and solid lines represent the similarity measured at the start and end of the training, respectively. The shaded area represents the maximum and minimum similarities over training. Residual connections with deep layers introduce gradient expansion.

$10^{11}$. When both similarities are close to zero, then the correlation is weak, and we cannot estimate one from the other. It is worth noting that the high gradient similarity can only be observed from the models with residual connections, corroborating the findings in Theorem 2. Similarly, the node representation expansion occurs with the residual connections.

If two similarities are not the same, which better represents the model performance on the test set? In Figure 3(b,c), we plot the relationship between each measure and test accuracy to answer this question. In Figure 3(b), we observe that when the model achieves relatively good test accuracy, the representation similarity ranges between $10^2$ and $10^{11}$. Since this range is quite broad, cases with low accuracy are also observed in this area. In addition, test accuracy is generally relatively low when the node similarity is too high or too low. To compare, we find that the range of gradient similarity is narrow when the model performs well in Figure 3(c). Specifically, the gradient similarity ranges between $10^{-2}$ and 1 when the test performance is relatively high. More importantly, there is no case with relatively low performance in this range, showing that the gradient similarity is a more reliable estimate of the test performance. We provide the results on the CiteSeer and Chameleon dataset in Appendix E, where a similar trend was observed.

## 5.2 HOW DOES THE GRADIENT SIMILARITY CHANGE OVER TRAINING?

We investigate how the gradient similarity changes over the training to check whether the optimization can mitigate the gradient expansion over time. With the similar experimental settings used in Section 5.1, we plot how the gradient similarity changes with different models and layers in Figures 4 and 5. We apply early stopping for all experiments.

Figure 4 shows the changes in gradient similarity without residual connections. For all experiments, the similarity increases over training, showing that some level of smoothing happens at the early stage of training. The results show that mild oversmoothing with 16 layers can be overcome through

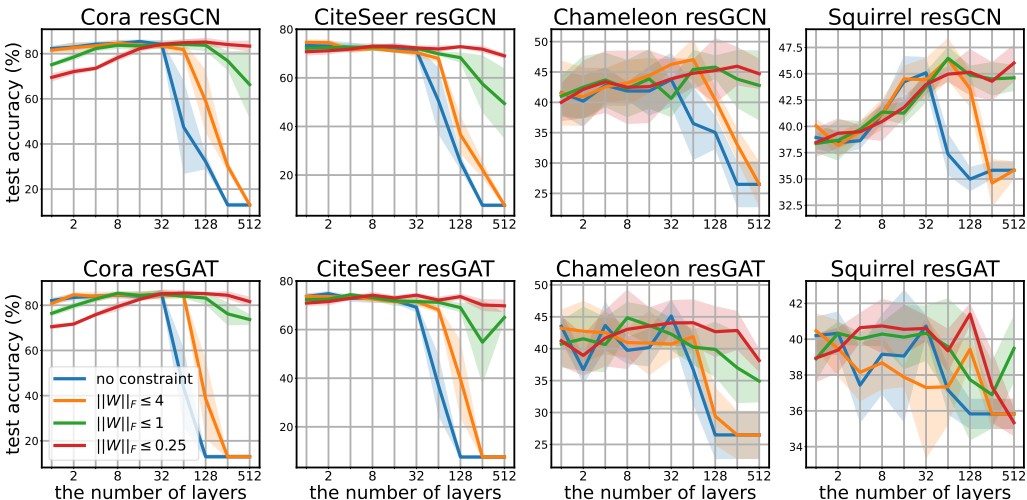

Figure 6: Performance of resGCN and resGAT on Cora, CiteSeer, Chameleon and Squirrel datasets with varying coefficient values. The solid curve indicates the average over five runs and the shaded area represents one standard deviation around the average.

training, but hard oversmoothing with 64 layers cannot be overcome for both models. Figure 5 shows the changes in gradient similarity with the residual connections. The gradient expansion can be observed with 64-layered models. Similar to the oversmoothing, once the expansion happens, the training process cannot be done properly (cf., Figure 3).

## 6 EFFECTIVENESS OF LIPTSCHITZ UPPER BOUND ON DEEP GNNs

We propose a Lipschitz upper bound constraint on each layer of GNN through the weight normalization in Section 4.1. In this section, we show the effectiveness of a Lipschitz upper bound constraint in improving test performance and training behavior in deep GNNs.

To evaluate test performance, we train resGCN and resGAT on Cora, CiteSeer, Chameleon, and Squirrel datasets with varying numbers of layers from 1 to 512. For the weight normalization, we test three different Lipschitz upper bound $c$: 4, 1, 0.25. All experiments are conducted with early stopping and learning rates of $0.001, 0.005, 0.01$. We report average values of five repeated runs.

The test accuracy of resGCN is reported in Figure 6 with and without weight normalization. For all datasets, we can observe a significant performance drop when the depth exceeds 32 without the weight normalization. Similar patterns are observed when a loose Lipschitz upper bound of $c = 4$ is applied. With the tight upper bounds of $c = 1, 0.25$, the test accuracy remains relatively consistent over 512 layers. Specifically, when $c = 0.25$, the accuracy of resGCN consistently improves over a hundred layers, reaching the best performance at a depth of 512 layers on Squirrel datasets. However, small upper bounds can limit the expressive power of each layer. As a consequence, test accuracy with a small upper bound requires more layers to reach a similar level of accuracy that can be achieved by a small number of layers without weight normalization.

In the case of homophilic datasets, despite the multiple layers, the highest accuracy is similar regardless of the upper bound constraints. For example, resGCN without normalization achieves the best performance of $85.43$ when the number of layers is 16. In contrast, resGCN with an upper bound of 0.25 achieves the best performance of $85.14$ when the number of layers is 128. However, in heterophilic datasets, deep models with weight normalization outperform the models without the upper bound constraints. For instance, in the Chameleon dataset, without normalization, the best accuracy of $43.75$ is achieved at a depth of 32. In contrast, the best accuracy of $47.04$ is achieved at a depth of 64 with an upper bound of 4. It is claimed that the heterophilic node classification requires capturing long-range dependencies via multiple layers (Rusch et al., 2023a). Our results also support this claim with carefully designed multi-layered GNNs.

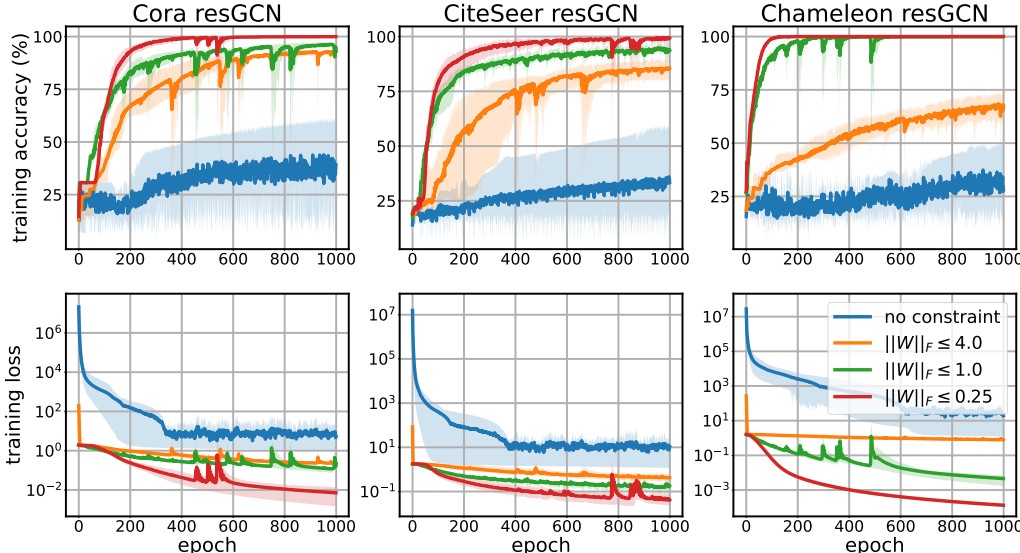

Figure 7: Training curves with Lipschitz upper bound constraint applied 64-layer resGCN over three datasets: Cora, Citeseer and Chameleon with learning rate $0.001$. The solid line indicates the average over five runs and the shaded area represents maximum and minimum value.

To demonstrate the training process, we train 64—and 128—layer resGCN and resGAT on Cora, CiteSeer, and Chameleon datasets with $1,000$ epochs and a fixed learning rate of $0.001$. For the weight normalization, we test three different Lipschitz upper bound $c$: $4, 1, 0.25$. The visualized training accuracy and loss on 64-layered resGCN are provided in Figure 7. We observe that applying the Lipschitz upper bound mitigates the underfitting problem of resGCN in all datasets. Specifically, resGCN with $c = 0.25$ achieves $100\%$ training accuracy in all datasets with the fastest convergence rate. Our results show that applying the Lipschitz upper bound effectively resolves the gradient expansion problem and helps optimization. We provide the results for 128—layer resGCN, and 64—and 128—layer resGAT in Appendix F.

## 7 CONCLUSION

In this work, we explored the challenges associated with optimizing deep GNNs. Until now, the oversmoothing has focused on the representation space. However, we demonstrate that oversmoothing also occurs in the gradient space, a major cause making the training process complicated. We further analyze the effect of the residual connections in the gradients of deepGNNs, revealing that the residual connection causes gradient expansion. To alleviate the gradient expansion during training, we constrain the Lipschitz upper bound on each layer through the Frobenius normalization. Experimental results show that the training procedure can be stabilized with the normalization over several hundred layers.

**Limitation.** The theoretical analysis is based on linear GCNs. Although we have shown that similar patterns are observable with non-linear GCNs and GATs through empirical studies, the non-linearity can introduce subtle differences compared with our analysis. We propose a weight normalization to restrict the Lipschitz constant. There are, however, several normalization approaches, such as the batch normalization (Ioffe & Szegedy, 2015) and the layer normalization (Ba, 2016). In a previous study, these normalizations are not helpful in deepGNNs (Chen et al., 2022). To investigate the reason further, we need to study the gradients of these normalizations in GNNs. We leave this future work.

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

## A MISSING PROOFS IN SECTION 4.1

In this section, we provide missing proofs in Section 4.1.

### A.1 PROOF OF LEMMA 1

*Proof.* We recap the derivation of the gradient of LGN, which is originally shown in Blakely et al. (2021). Using the chain rule, we can compute:

$$
\begin{aligned}
\frac{\partial \mathcal{L}_{\text{LGN}}}{\partial \mathbf{W}^{(\ell)}} &= \frac{\partial \mathcal{L}_{\text{LGN}}}{\partial \mathbf{X}^{(\ell+1)}} \frac{\partial \mathbf{X}^{(\ell+1)}}{\partial \mathbf{W}^{(\ell)}} \\
&= \frac{\partial \mathcal{L}_{\text{LGN}}}{\partial \mathbf{X}^{(\ell+1)}} \frac{\partial}{\partial \mathbf{W}^{(\ell)}} \left( \hat{\mathbf{A}} \mathbf{X}^{(\ell)} \mathbf{W}^{(\ell)} \right) \\
&= (\hat{\mathbf{A}} \mathbf{X}^{(\ell)})^{\top} \frac{\partial \mathcal{L}_{\text{LGN}}}{\partial \mathbf{X}^{(\ell+1)}} \,.
\end{aligned}
$$

Again, by using the chain rule, we have:

$$
\begin{aligned}
\frac{\partial \mathcal{L}_{\mathsf{LGN}}}{\partial \mathbf{X}^{(\ell)}} &= \frac{\partial \mathcal{L}_{\mathsf{LGN}}}{\partial \mathbf{X}^{(L)}} \frac{\partial \mathbf{X}^{(L)}}{\partial \mathbf{X}^{(L-1)}} \cdots \frac{\partial \mathbf{X}^{(\ell+2)}}{\partial \mathbf{X}^{(\ell+1)}} \frac{\partial \mathbf{X}^{(\ell+1)}}{\partial \mathbf{X}^{(\ell)}} \\
&= \left( \frac{\partial \mathcal{L}_{\mathsf{LGN}}}{\partial \mathbf{X}^{(L)}} \frac{\partial \mathbf{X}^{(L)}}{\partial \mathbf{X}^{(L-1)}} \cdots \frac{\partial \mathbf{X}^{(\ell+2)}}{\partial \mathbf{X}^{(\ell+1)}} \right) \frac{\partial}{\partial \mathbf{X}^{(\ell)}} \hat{\mathbf{A}} \mathbf{X}^{(\ell)} \mathbf{W}^{(\ell)} \\
&= \hat{\mathbf{A}}^{\top} \left( \frac{\partial \mathcal{L}_{\mathsf{LGN}}}{\partial \mathbf{X}^{(L)}} \frac{\partial \mathbf{X}^{(L)}}{\partial \mathbf{X}^{(L-1)}} \cdots \frac{\partial \mathbf{X}^{(\ell+2)}}{\partial \mathbf{X}^{(\ell+1)}} \right) (\mathbf{W}^{(\ell)})^{\top} \\
&= (\hat{\mathbf{A}}^{\top})^2 \left( \frac{\partial \mathcal{L}_{\mathsf{LGN}}}{\partial \mathbf{X}^{(L)}} \frac{\partial \mathbf{X}^{(L)}}{\partial \mathbf{X}^{(L-1)}} \cdots \frac{\partial \mathbf{X}^{(\ell+3)}}{\partial \mathbf{X}^{(\ell+2)}} \right) (\mathbf{W}^{(\ell+1)})^{\top} (\mathbf{W}^{(\ell)})^{\top} \\
&= \cdots \\
&= (\hat{\mathbf{A}}^{\top})^{L-\ell} \frac{\partial \mathcal{L}_{\mathsf{LGN}}}{\partial \mathbf{X}^{(L)}} \prod_{i=1}^{L-\ell} (\mathbf{W}^{(L-i)})^{\top} .
\end{aligned}
$$

$\square$

### A.2  PROOF OF THEOREM 1

*Proof.*  To prove Theorem 1, we recap and use the following definitions and lemmas from Wu et al. (2023a).

**Definition 1** (Ergodicity). Let $\mathbf{B} \in \mathbb{R}^{(N-1) \times N}$ be the orthogonal projection onto the space orthogonal to $\mathsf{span}\{\mathbf{1}\}$. A sequence of matrices $\{M^{(n)}\}_{n=1}^{\infty}$ is ergodic if

$$
\lim_{t \to \infty} \mathbf{B} \prod_{n=0}^{t} \mathbf{M}^{(n)} = 0.
$$

By using the projection matrix $\mathbf{B}$, we can express $\mu(\mathbf{X})$ as $\|\mathbf{B}\mathbf{X}\|_F$. We want to show that $\{\hat{\mathbf{A}}\}_{n=1}^{\infty}$ is ergodic. Since $\hat{\mathbf{A}} = \tilde{\mathbf{D}}^{-\frac{1}{2}} \tilde{\mathbf{A}} \tilde{\mathbf{D}}^{-\frac{1}{2}}$ and $\tilde{\mathbf{D}}^{-1} \tilde{\mathbf{A}}$ have a same spectrum, $\{\hat{\mathbf{A}}\}_{n=1}^{\infty}$ is ergodic if $\{\tilde{\mathbf{D}}^{-1} \tilde{\mathbf{A}}\}_{n=1}^{\infty}$ is ergodic. Using Lemma 3 in (Wu et al., 2023a), $\{\tilde{\mathbf{D}}^{-1} \tilde{\mathbf{A}}\}_{n=1}^{\infty}$ is ergodic.

Next, we need notion of joint spectral radius to show the convergence rate.

**Definition 2** (Joint Spectral Radius). For a collection of matrices $\mathcal{A}$, the joint spectral radius $\mathsf{JSR}(\mathcal{A})$ is defined to be

$$
\mathsf{JSR}(\mathcal{A}) = \limsup_{k \to \infty} \sup_{\mathbf{A}_1, \mathbf{A}_2, \dots, \mathbf{A}_k \in \mathcal{A}} \|\mathbf{A}_1 \mathbf{A}_2 \cdots \mathbf{A}_k\|^{\frac{1}{k}}
$$

and it is independent of the norm used.

By the definition, it is straight forward that if $\mathsf{JSR}(\mathcal{A}) < 1$, for any $\mathsf{JSR}(\mathcal{A}) < q < 1$, there exists a $C$ for which satisfies $\|\mathbf{A}_1 \mathbf{A}_2 \cdots \mathbf{A}_k \mathbf{y}\| \leq C q^k \|\mathbf{y}\|$ for $\mathbf{A}_1, \mathbf{A}_2, \cdots, \mathbf{A}_k \in \mathcal{A}$. Let $\hat{\mathcal{A}} = \{\hat{\mathbf{A}}\}$ and $\tilde{\mathcal{A}} = \{\tilde{\mathbf{D}}^{-1} \tilde{\mathbf{A}}\}$. Similar to Ergodicity, we can get $\mathsf{JSR}(\hat{\mathcal{A}}) < 1$ if $\mathsf{JSR}(\tilde{\mathcal{A}}) < 1$. Using Lemma 6 in (Wu et al., 2023a), $\mathsf{JSR}(\tilde{\mathcal{A}}) < 1$.

Now, we can derive:

$$
\begin{aligned}
\mu \left( \frac{\partial \mathcal{L}_{\mathsf{LGN}}}{\partial \mathbf{X}^{(\ell)}} \right) &= \|\mathbf{B} \frac{\partial \mathcal{L}_{\mathsf{LGN}}}{\partial \mathbf{X}^{(\ell)}}\|_F \\
&= \|\mathbf{B} (\hat{\mathbf{A}}^{\top})^{L-\ell} \frac{\partial \mathcal{L}_{\mathsf{LGN}}}{\partial \mathbf{X}^{(L)}} \prod_{i=1}^{L-\ell} (\mathbf{W}^{(L-i)})^{\top}\|_F \\
&\leq \|\frac{\partial \mathcal{L}_{\mathsf{LGN}}}{\partial \mathbf{X}^{(L)}}\|_F \|\mathbf{B}(\hat{\mathbf{A}}^{\top})^{L-\ell}\|_2 \|\mathbf{W}^{(*)}\|_2^{L-\ell} ,
\end{aligned}
$$

where $\|\mathbf{W}^{(*)}\|_2$ is the maximum spectral norm, i.e., $* = \arg\max_i \|\mathbf{W}^{(i)}\|_2$ for $1 \leq i \leq L - \ell$. By using $\mathsf{JSR}(\hat{\mathcal{A}}) < 1$, we can result in to Theorem 1.

$\square$

### A.3 PROOF OF LEMMA 2

*Proof.* By applying the chain rule, we can obtain:

$$
\begin{aligned}
\frac{\partial \mathcal{L}_{\text{resLGN}}}{\partial \mathbf{W}^{(\ell)}} &= \frac{\partial \mathcal{L}_{\text{resLGN}}}{\partial \mathbf{X}^{(\ell+1)}} \frac{\partial \mathbf{X}^{(\ell+1)}}{\partial \mathbf{W}^{(\ell)}} \\
&= \frac{\partial \mathcal{L}_{\text{resLGN}}}{\partial \mathbf{X}^{(\ell+1)}} \frac{\partial}{\partial \mathbf{W}^{(\ell)}} \left( \mathbf{X}^{(\ell)} + \hat{\mathbf{A}} \mathbf{X}^{(\ell)} \mathbf{W}^{(\ell)} \right) \\
&= \left( \hat{\mathbf{A}} \mathbf{X}^{(\ell)} \right)^{\top} \frac{\partial \mathcal{L}_{\text{resLGN}}}{\partial \mathbf{X}^{(\ell+1)}} .
\end{aligned}
$$

We have:

$$
\frac{\partial \mathcal{L}_{\text{resLGN}}}{\partial \mathbf{X}^{(\ell)}} = \frac{\partial \mathcal{L}_{\text{resLGN}}}{\partial \mathbf{X}^{(L)}} \frac{\partial \mathbf{X}^{(L)}}{\partial \mathbf{X}^{(L-1)}} \frac{\partial \mathbf{X}^{(L-1)}}{\partial \mathbf{X}^{(L-2)}} \cdots \frac{\partial \mathbf{X}^{(\ell+1)}}{\partial \mathbf{X}^{(\ell)}} . \tag{7}
$$

We will show Equation (5) holds inductively. First two components of Equation (7) can be derived as follows:

$$
\begin{aligned}
\frac{\partial \mathcal{L}_{\text{resLGN}}}{\partial \mathbf{X}^{(L)}} \frac{\partial \mathbf{X}^{(L)}}{\partial \mathbf{X}^{(L-1)}} &= \frac{\partial \mathcal{L}_{\text{resLGN}}}{\partial \mathbf{X}^{(L)}} \frac{\partial}{\partial \mathbf{X}^{(L-1)}} (\mathbf{X}^{(L-1)} + \hat{\mathbf{A}} \mathbf{X}^{(L-1)} \mathbf{W}^{(L-1)}) \\
&= \frac{\partial \mathcal{L}_{\text{resLGN}}}{\partial \mathbf{X}^{(L)}} + (\hat{\mathbf{A}})^{\top} \frac{\partial \mathcal{L}_{\text{resLGN}}}{\partial \mathbf{X}^{(L)}} (\mathbf{W}^{(L-1)})^{\top} ,
\end{aligned} \tag{8}
$$

which satisfies Equation (5) with $\ell = L - 1$. Suppose we have $\frac{\partial \mathcal{L}_{\text{resLGN}}}{\partial \mathbf{X}^{(\ell+1)}}$ which satisfies Equation (5). We can derive:

$$
\begin{aligned}
\frac{\partial \mathcal{L}_{\text{resLGN}}}{\partial \mathbf{X}^{(\ell)}} =& \frac{\partial \mathcal{L}_{\text{resLGN}}}{\partial \mathbf{X}^{(\ell+1)}} \frac{\partial \mathbf{X}^{(\ell+1)}}{\partial \mathbf{X}^{(\ell)}} \\
=& \frac{\partial \mathcal{L}_{\text{resLGN}}}{\partial \mathbf{X}^{(\ell+1)}} \frac{\partial}{\partial \mathbf{X}^{(\ell)}} (\mathbf{X}^{(\ell)} + \hat{\mathbf{A}} \mathbf{X}^{(\ell)} \mathbf{W}^{(\ell)}) \\
=& \frac{\partial \mathcal{L}_{\text{resLGN}}}{\partial \mathbf{X}^{(\ell+1)}} + \hat{\mathbf{A}}^{\top} \frac{\partial \mathcal{L}_{\text{resLGN}}}{\partial \mathbf{X}^{(\ell+1)}} (\mathbf{W}^{(\ell)})^{\top} \\
=& \frac{\partial \mathcal{L}_{\text{resLGN}}}{\partial \mathbf{X}^{(L)}} + \hat{\mathbf{A}}^{\top} \frac{\partial \mathcal{L}_{\text{resLGN}}}{\partial \mathbf{X}^{(L)}} (\mathbf{W}^{(\ell)})^{\top} \\
&+ \sum_{p=1}^{L-\ell-1} \left( \sum_{\{i_1 : i_1 \geq \ell+1\}} \cdots \sum_{\{i_p : i_{p-1} < i_p < L\}} \left( \hat{\mathbf{A}}^{\top} \right)^{p} \frac{\partial \mathcal{L}_{\text{resLGN}}}{\partial \mathbf{X}^{(L)}} \prod_{k=0}^{p-1} \left( \mathbf{W}^{(i_{p-k})} \right)^{\top} \right) \\
&+ \sum_{p=1}^{L-\ell-1} \left( \sum_{\{i_1 : i_1 \geq \ell+1\}} \cdots \sum_{\{i_p : i_{p-1} < i_p < L\}} \left( \hat{\mathbf{A}}^{\top} \right)^{p+1} \frac{\partial \mathcal{L}_{\text{resLGN}}}{\partial \mathbf{X}^{(L)}} \left( \prod_{k=0}^{p-1} \left( \mathbf{W}^{(i_{p-k})} \right)^{\top} \right) (\mathbf{W}^{(\ell)})^{\top} \right) \\
=& \frac{\partial \mathcal{L}_{\text{resLGN}}}{\partial \mathbf{X}^{(L)}} + \hat{\mathbf{A}}^{\top} \frac{\partial \mathcal{L}_{\text{resLGN}}}{\partial \mathbf{X}^{(L)}} (\mathbf{W}^{(\ell)})^{\top} + \sum_{\ell+1 \leq i_1 < L} \hat{\mathbf{A}}^{\top} \frac{\partial \mathcal{L}_{\text{resLGN}}}{\partial \mathbf{X}^{(L)}} (\mathbf{W}^{(i_1)})^{\top} \\
&+ \sum_{p=2}^{L-\ell-1} \left( \sum_{\{i_1 : i_1 \geq \ell+1\}} \cdots \sum_{\{i_p : i_{p-1} < i_p < L\}} \left( \hat{\mathbf{A}}^{\top} \right)^{p} \frac{\partial \mathcal{L}_{\text{resLGN}}}{\partial \mathbf{X}^{(L)}} \prod_{k=0}^{p-1} \left( \mathbf{W}^{(i_{p-k})} \right)^{\top} \right) \\
&+ \sum_{p=1}^{L-\ell-2} \left( \sum_{\{i_1 : i_1 \geq \ell+1\}} \cdots \sum_{\{i_p : i_{p-1} < i_p < L\}} \left( \hat{\mathbf{A}}^{\top} \right)^{p+1} \frac{\partial \mathcal{L}_{\text{resLGN}}}{\partial \mathbf{X}^{(L)}} \left( \prod_{k=0}^{p-1} \left( \mathbf{W}^{(i_{p-k})} \right)^{\top} \right) (\mathbf{W}^{(\ell)})^{\top} \right) \\
&+ \left( \hat{\mathbf{A}}^{\top} \right)^{L-\ell} \frac{\partial \mathcal{L}_{\text{resLGN}}}{\partial \mathbf{X}^{(L)}} \left( \prod_{i=1}^{L-\ell} (\mathbf{W}^{(L-i)})^{\top} \right) .
\end{aligned}
$$

The fourth term and the fifth term can be merged into:

$$
\sum_{p=2}^{L-\ell-1} \left( \sum_{\{i_1 : i_1 \geq \ell\}} \cdots \sum_{\{i_p : i_{p-1} < i_p < L\}} \left( \hat{\mathbf{A}}^{\top} \right)^{p} \frac{\partial \mathcal{L}_{\text{resLGN}}}{\partial \mathbf{X}^{(L)}} \prod_{k=0}^{p-1} \left( \mathbf{W}^{(i_{p-k})} \right)^{\top} \right) ,
$$

since $\binom{L-\ell-1}{p+1} + \binom{L-\ell-1}{p} = \binom{L-\ell}{p+1}$. Therefore,

$$\frac{\partial \mathcal{L}_{\text{resLGN}}}{\partial \mathbf{X}^{(\ell)}} = \frac{\partial \mathcal{L}_{\text{resLGN}}}{\partial \mathbf{X}^{(L)}} + \sum_{\ell \leq i_1 < L} \hat{\mathbf{A}}^\top \frac{\partial \mathcal{L}_{\text{resLGN}}}{\partial \mathbf{X}^{(L)}} (\mathbf{W}^{(i_1)})^\top$$

$$+ \sum_{p=2}^{L-\ell-1} \left( \sum_{\{i_1 : i_1 \geq \ell\}} \cdots \sum_{\{i_p : i_{p-1} < i_p < L\}} \left( \hat{\mathbf{A}}^\top \right)^p \frac{\partial \mathcal{L}_{\text{resLGN}}}{\partial \mathbf{X}^{(L)}} \prod_{k=0}^{p-1} \left( \mathbf{W}^{(i_{p-k})} \right)^\top \right)$$

$$+ \left( \hat{\mathbf{A}}^\top \right)^{L-\ell} \frac{\partial \mathcal{L}_{\text{resLGN}}}{\partial \mathbf{X}^{(L)}} \left( \prod_{i=1}^{L-\ell} (\mathbf{W}^{(L-i)})^\top \right)$$

$$= \frac{\partial \mathcal{L}_{\text{resLGN}}}{\partial \mathbf{X}^{(L)}} + \sum_{p=1}^{L-\ell} \left( \sum_{\{i_1 : i_1 \geq \ell\}} \cdots \sum_{\{i_p : i_{p-1} < i_p < L\}} \left( \hat{\mathbf{A}}^\top \right)^p \frac{\partial \mathcal{L}_{\text{resLGN}}}{\partial \mathbf{X}^{(L)}} \prod_{k=0}^{p-1} \left( \mathbf{W}^{(i_{p-k})} \right)^\top \right),$$

which satisfies Equation (5). $\qquad\square$

### A.4 PROOF OF THEOREM 2

*Proof.* We can obtain:

$$\mu \left( \frac{\partial \mathcal{L}_{\text{resGNN}}(\mathbf{W})}{\partial \mathbf{X}^{(\ell)}} \right) = \| \mathbf{B} \frac{\partial \mathcal{L}_{\text{resGNN}}(\mathbf{W})}{\partial \mathbf{X}^{(\ell)}} \|_F$$

$$\leq \| \mathbf{B} \frac{\partial \mathcal{L}_{\text{resLGN}}}{\partial \mathbf{X}^{(L)}} \|_F$$

$$+ \sum_{p=1}^{L-\ell} \sum_{\{i_1 : i_1 \geq \ell\}} \cdots \sum_{\{i_p : i_{p-1} < i_p < L\}} \| \mathbf{B} \left( \hat{\mathbf{A}}^\top \right)^p \frac{\partial \mathcal{L}_{\text{resLGN}}}{\partial \mathbf{X}^{(L)}} \prod_{k=0}^{p-1} \left( \mathbf{W}^{(i_{p-k})} \right)^\top \|_F .$$

Since there are $\binom{L-\ell}{p}$ sets of $(i_1, ... i_p)$ which satisfies $\ell \leq i_1 < i_2 < \cdots < i_p < L$,

$$\sum_{\{i_1 : i_1 \geq \ell\}} \cdots \sum_{\{i_p : i_{p-1} < i_p < L\}} \| \mathbf{B} \left( \hat{\mathbf{A}}^\top \right)^p \frac{\partial \mathcal{L}_{\text{resLGN}}}{\partial \mathbf{X}^{(L)}} \prod_{k=0}^{p-1} \left( \mathbf{W}^{(i_{p-k})} \right)^\top \|_F$$

$$\leq \binom{L-\ell}{p} \| \frac{\partial \mathcal{L}_{\text{LGN}}}{\partial \mathbf{X}^{(L)}} \|_F \| \mathbf{B}(\hat{\mathbf{A}}^\top)^{L-\ell} \|_2 \| \mathbf{W}^{(*)} \|_2^{L-\ell} ,$$

where $\| \mathbf{W}^{(*)} \|_2$ is the maximum spectral norm, i.e., $* = \arg\max_i \| \mathbf{W}^{(i)} \|_2$ for $1 \leq i \leq L - \ell$. By using $\text{JSR}(\hat{\mathcal{A}}) < 1$, there exists $0 < q < 1$ and $C_q > 0$ such that

$$\mu \left( \frac{\partial \mathcal{L}_{\text{resGNN}}(\mathbf{W})}{\partial \mathbf{X}^{(\ell)}} \right) \leq \mu \left( \mathbf{B} \frac{\partial \mathcal{L}_{\text{resLGN}}}{\partial \mathbf{X}^{(L)}} \right) + \sum_{p=1}^{L-\ell} C_q \binom{L-\ell}{p} (q \| \mathbf{W}^{(*)} \|_2)^{L-\ell}$$

$$= \mu \left( \mathbf{B} \frac{\partial \mathcal{L}_{\text{resLGN}}}{\partial \mathbf{X}^{(L)}} \right) + C_q \left( \left( 1 + q \| \mathbf{W}^{(*)} \|_2 \right)^{L-\ell} - 1 \right) .$$

$\qquad\square$

## B DATASET STATISTICS

Statistics of datasets used in our experiments can be found in Table 1. We note that the filtering process proposed by Platonov et al. (2023) was adopted in the case of Chameleon and Squirrel datasets.

## C SUPPLEMENTARY VISUALIZATIONS OF GRADIENT SIMILARITY EVALUATIONS

Supplementary visualizations for the analysis in Section 4.2 are provided below. We did not plot gradient similarity measure of 128-layer GAT, as the measures yielded "NaN" values in all layers except on last layer because of expansion.

| Dataset | Type | # nodes | # edges | # classes | avg degree |
|---------|------|---------|---------|-----------|------------|
| Cora | homophilic | 2,708 | 5,278 | 7 | 3.90 |
| CiteSeer | homophilic | 3,327 | 4,552 | 6 | 2.74 |
| Squirrel | heterophilic | 2,223 | 46,998 | 5 | 42.28 |
| Chameleon | heterophilic | 890 | 8,854 | 5 | 19.90 |

Table 1: Statistics of the dataset utilized in the experiments.

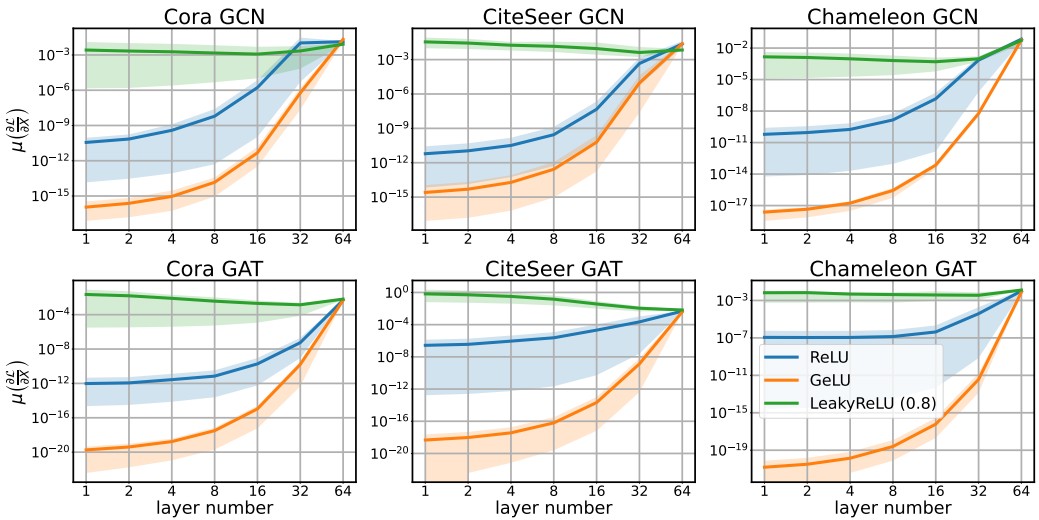

Figure 8: Gradient similarity measure $\mu\left(\frac{\partial \mathcal{L}_{\text{GNN}}(\mathbf{W})}{\partial \mathbf{X}^{(\ell)}}\right)$ over different layers and activation functions. We report the oversmoothing measures of 64-layer GCN and GAT over three datasets: Cora, Cite-Seer, and Chameleon.

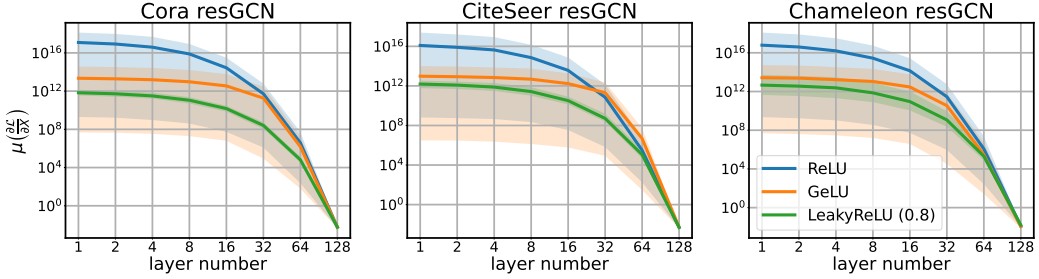

Figure 9: Gradient similarity measure $\mu\left(\frac{\partial \mathcal{L}_{\text{resGNN}}(\mathbf{W})}{\partial \mathbf{X}^{(\ell)}}\right)$ over different layers of 128-layer GCN with residual connections.

## D  VISUALIZATIONS OF TRAINING CURVES

Supplementary visualizations for the 4-, 16-, 64- layer GCN, GAT, resGCN and resGAT are provided below.

## E  SUPPLEMENTARY SCATTER PLOTS

Scatter plots for the analysis in Section 5.1 are provided below.

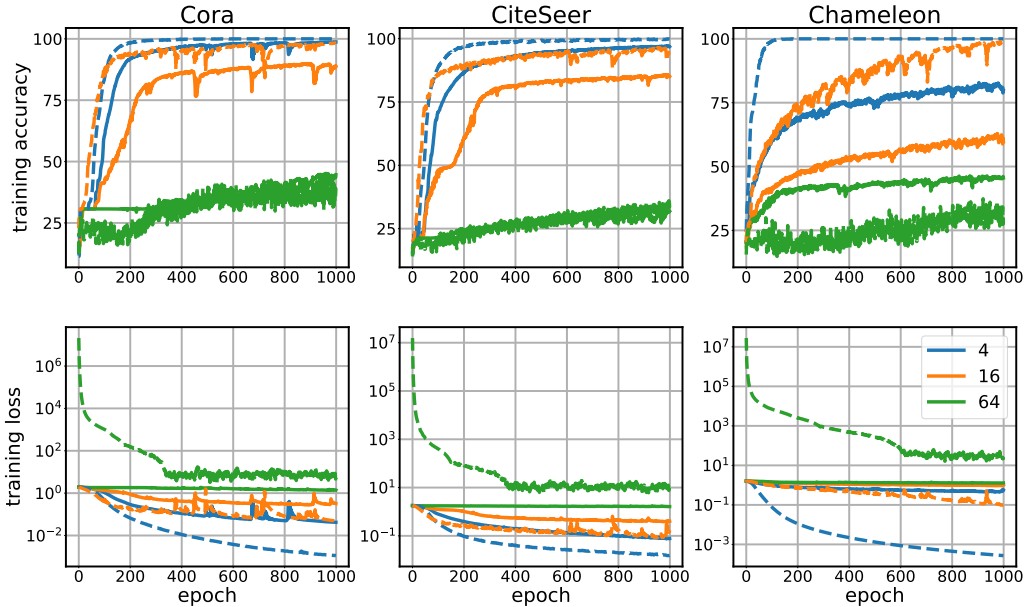

Figure 10: Training curves 4-, 16-, 64- layer GCN and resGCN over three datasets: Cora, Citeseer, and Chameleon with learning rate 0.001. The dashed lines represent the existence of residual connection while solid lines represent the vanilla model.

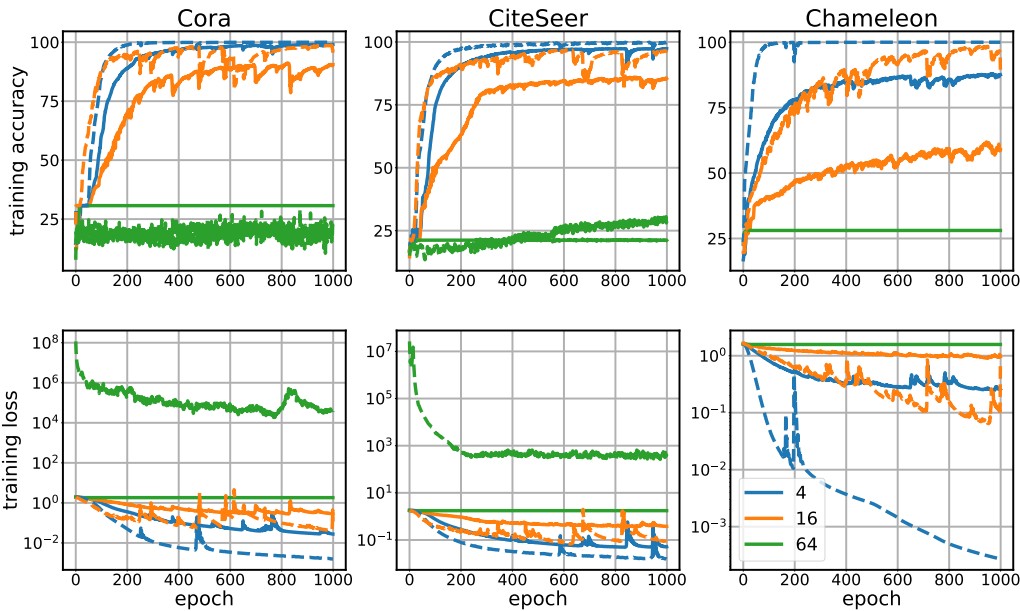

Figure 11: Training curves 4-, 16-, 64- layer GAT and resGAT over three datasets: Cora, Citeseer, and Chameleon with learning rate 0.001. The dashed lines represent the existence of residual connection, while solid lines represent the vanilla model.

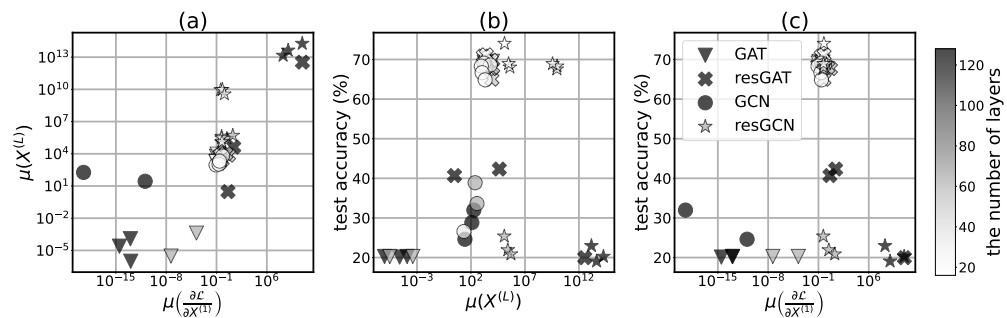

Figure 12: Scatter plots between (a) representation similarity vs gradient similarity (b) test accuracy and representation similarity, (c) test accuracy and gradient similarity on the CiteSeer dataset.

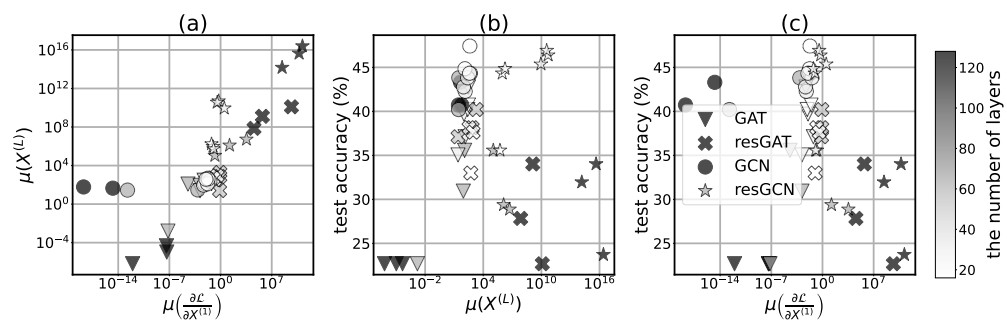

Figure 13: Scatter plots between (a) representation similarity vs gradient similarity (b) test accuracy and representation similarity, (c) test accuracy and gradient similarity on the Chameleon dataset.

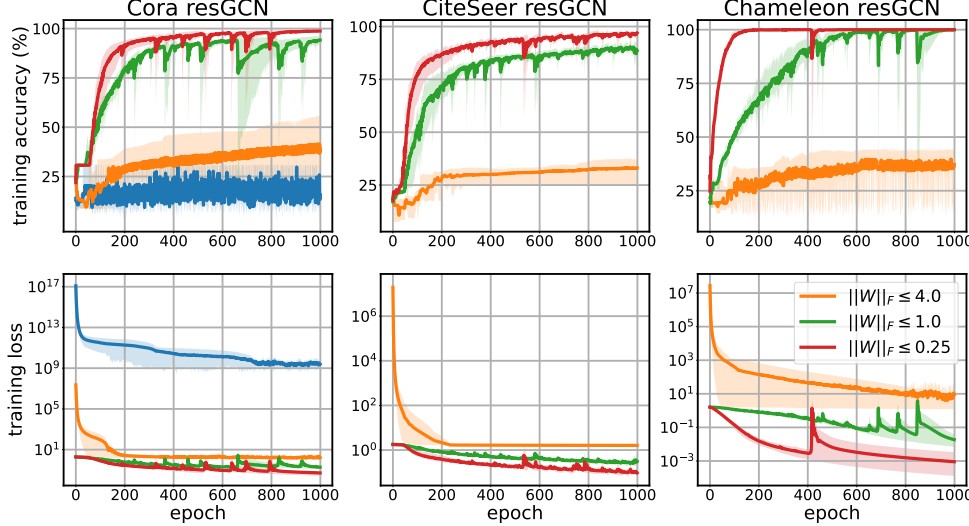

Figure 14: Training curves with Lipschitz upper bound constraint applied 128-layer resGCN over three datasets: Cora, Citeseer and Chameleon with learning rate $0.001$.

## F    SUPPLEMENTARY VISUALIZATIONS OF TRAINING CURVES WITH LIPSCHITZ UPPER BOUND CONSTRAINT

Supplementary visualizations of training curves with Lipschitz upper bound constraint for Section 6 are provided below.

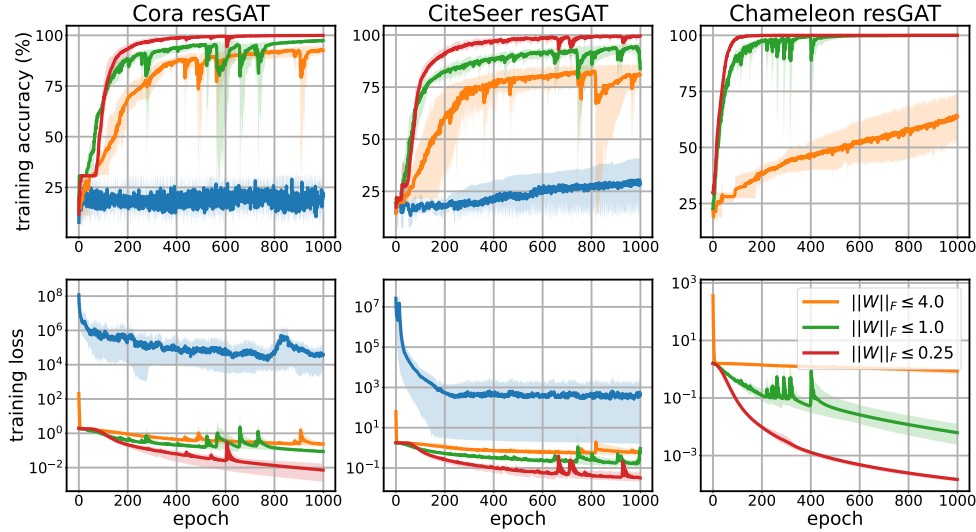

Figure 15: Training curves with Lipschitz upper bound constraint applied $64$-layer resGAT over three datasets: Cora, Citeseer and Chameleon with learning rate $0.001$.

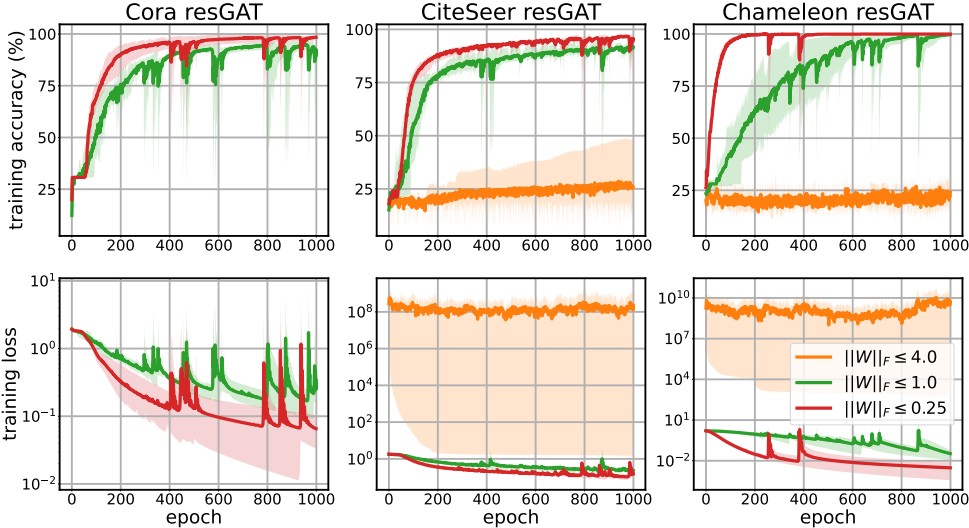

Figure 16: Training curves with Lipschitz upper bound constraint applied $128$-layer resGAT over three datasets: Cora, Citeseer and Chameleon with learning rate $0.001$.

