# OpenReview forum: "Taming Gradient Oversmoothing and Expansion in Graph Neural Networks"
_ICLR.cc/2025/Conference — ICLR 2025 Conference Withdrawn Submission_

### Official Review · Reviewer_nbzZ · 2024-10-28

**Soundness:** 3
**Presentation:** 3
**Contribution:** 2
**Rating:** 5
**Confidence:** 4

**Summary:**

The paper analyzed the over-smoothing phenomenon in Graph Neural Networks (GNNs) from a unique perspective, uncovering the issue of gradient expansion in residual connections. To address the problem of gradient explosion, the author introduced a layer-wise Lipschitz constraint, which facilitates efficient training of residual GNNs and enhances their performance.

**Strengths:**

1. The paper presents a comprehensive theoretical framework for analyzing the over-smoothing phenomenon in GNNs and gradient expansion in residual GNNs, introducing the concepts of representation similarity and gradient similarity to further explain these phenomena.

2. The mathematical derivations are rigorous yet easy to comprehend.

3. The author extensively investigates gradient similarity and representation similarity through numerous experiments, examining their variation throughout the training process.

4. The paper proposes a Lipschitz constraint method that effectively alleviates the effects of over-smoothing and gradient expansion during the training process.

**Weaknesses:**

1. In line 147, the equation $$ \||\mathbf{X}\||_F^2 = \mu(\mathbf{X})^2 + \||\mathbf{X} - \mathbf{B}\mathbf{X}\||_F^2 $$ is presented. a step-by-step derivation or proof of this equation in an appendix, which will greatly enhance the clarity of the methodology presented.

2. The authors provided a mathematical derivation for representation similarity and gradient similarity, validating their correctness in Figures 1 and 2. However, it is unclear which metric better measures over-smoothing, as the relationships among representation similarity, gradient similarity, and test accuracy appear ambiguous in Figure 3. It seems the authors primarily used gradient similarity to assess over-smoothing in Section 5. In references [1] and [2], Dirichlet energy appears to be a more effective metric compared to gradient similarity.  It would be helpful if  authors could provide a more detailed comparison between gradient similarity and other metrics like Dirichlet energy, including quantitative results, which would help clarify the advantages of their chosen metric.

3. In my view, large datasets can mitigate oversmoothing compared to smaller ones. Intuitively, when aggregating features, each node has more distant neighbors, making it harder for features to become overly similar. As shown in [2], Pubmed and Ogbn-arxiv are less affected, as seen in Table 1. Could authors extend their theoretical analysis to consider the impact of dataset size on oversmoothing to enhance the theoretical persuasiveness of the paper?

4. The Lipschitz constraint method is an effective approach to alleviating over-smoothing and gradient expansion. It would enhance the paper's strength to incorporate 2-3 specific baseline methods from the related work section and utilize 1-2 large datasets from references [1] and [2] that are particularly relevant for comparison.

5. It appears that the theoretical derivation in the paper is based on the GCN model, yet the authors have also conducted some experiments on GAT, such as in Figure 4. Perhaps the authors should provide a corresponding theoretical analysis to enhance the persuasiveness of the paper. Additionally, if the model is changed to another one, such as GraphSage, would the theory still be useful?

[1] A Survey on Oversmoothing in Graph Neural Networks
[2] Dirichlet Energy Constrained Learning for Deep Graph Neural Networks

**Questions:**

Refer to weaknesses

---

### Official Review · Reviewer_Yjtw · 2024-10-29

**Soundness:** 3
**Presentation:** 4
**Contribution:** 3
**Rating:** 5
**Confidence:** 4

**Summary:**

The authors analyze the effects of the addition of  residual connections in **linear** GNNs **with no non-linear activation function** (specifically GCNNs) as an effort to compact oversmoothing (in extremely deep GNNs).  In a pair of two results (Theorems 1 and 2) the authors analyize the decay of a (reasonable) orthogonal projection of the gradient of the loss with respect to pertrubations of entires of the feature matrix.

The use their theoretical upper bounds to suggest that spectral weight normalization, in the spirit of [1], would remedy oversmoothing.  Only empirical studies are then used to support at (presumably) theoretical claim.



[1] Neyshabur, Behnam, Russ R. Salakhutdinov, and Nati Srebro. "Path-sgd: Path-normalized optimization in deep neural networks." Advances in neural information processing systems 28 (2015).

**Strengths:**

The numerical experitments are very nicely executed, clear, convincing, and reproducible (but I still have questions as they do not seem to match the maths).  The bounds are reasonable (even if the proof is extremely straightforward....) and interesting.

**Weaknesses:**

- **Identity Activation Only**: Only identity activation are considered (it doesn't seem to be difficult to obtain a result for non-linear activation fixing the origin from your proofs....

- **Connected Graphs** Why not consider disconnected graphs?  It seems to me that with only a bit more work, you can generalize your result by noting that the normalized adjacency matrix has a block-diagonal form in the general (non-bipartide case).


- Results are for non-activated GNNs (identity activation function) but the numerics have non-linear activations (and result with identity activation are not plotted).  This latter point, namely that there are no numerical experiments matching the theoretical setting, makes me wonder if there is a gap...Does the theory really come through in practice.  One **needs** illustrations matching the architectural setup in the theorem.

- Theory is for very basic GCNN model (Which is definitely not universal by any means) but many other GNN models are considered in the experiments section.  What is the relevance?

**Questions:**

- As the graph size (number of nodes specifically) diverges, what happens to the proposed normalization?

- Do you not have any theoretical guarantees for the proposed procedure since you could not obtain matching lower bounds?  I assume that would be required to argue that the normalization does the trick.

- Can you add a few more words explaining what the trouble is with bipartide?

---

### Official Review · Reviewer_7mh1 · 2024-11-03

**Soundness:** 3
**Presentation:** 2
**Contribution:** 1
**Rating:** 3
**Confidence:** 4

**Summary:**

This paper studies the behavior of gradients in deep GNNs by considering the oversmoothing problem. The authors show that when GNNs are very deep, the representations become similar (which is a known thing), and that the gradients in first layers become very similar (makes sense by inspecting the Jacobian of the GNN).

The authors perform several experiments, mostly on simple node classification datasets, and show the training/test accuracies alongside the gradient and feature similarities.

**Strengths:**

In terms of strengths, the paper is nicely written and easy to follow. It also sheds more light on the oversmoothing problem. The experiments conducted to understand the theoretical findings are in-order.

**Weaknesses:**

In terms of relevance and novelty, as well as related works, I think that the main issue is that most of the presented results are already quite known within the community, and can be found in the literature. For example see "A survey on oversmoothing in graph neural networks"  and "Simplifying the Theory on Over-Smoothing". Also, the authors lack a discussion of "Revisiting Graph Neural Networks: All We Have is Low-Pass Filters".

In terms of using the Lipschitz constant and bounding it, it was shown in "On the Robustness of Graph Neural Diffusion to Topology Perturbations" and "Contractive Systems Improve Graph Neural Networks Against Adversarial Attacks" that using it can help to address robustness problems, so it would be interesting to understand the connection between oversmoothing and such approaches.

In terms of experiments, the authors used mostly simple datasets, and I think that it would be beneficial to also study the performance on other tasks and datasets. I think it would also be interesting to see a report of the Dirichlet energy, that is usually reported in oversmoothing studies. Also, I feel that the experiments lack comparisons with other methods that can address oversmoothing (there are already many such methods).

**Questions:**

Please see my suggestions in the weaknesses section.

---

### Official Review · Reviewer_m3nv · 2024-11-03

**Soundness:** 2
**Presentation:** 1
**Contribution:** 1
**Rating:** 3
**Confidence:** 4

**Summary:**

The paper argues that gradient oversmoothing and gradient expansion pose challenges in training deep Graph Neural Networks (GNNs). Unlike previous approaches that focus on node features, the authors apply a previously defined similarity measure to gradients. They also establish an asymptotic upper bound for this measure, describing the phenomenon where it approaches zero as "gradient oversmoothing," and linking its growth to "gradient expansion." To address these issues, the authors propose a novel normalization technique. Empirical results on various graph datasets and architectures demonstrate that this measure decreases as layer depth increases, and the new normalization approach enables successful training of deep GNNs.

**Strengths:**

The paper studies gradient behavior during training of deep GNNs. While this is a widely studied field, mainly linked to oversquashing these days, its full qualitative behavior remains to be studied in more detail. The paper is further a good combination of theoretical and empirical work.

**Weaknesses:**

My biggest concern is that the statement on gradient oversmoothing in the asymptotic case is likely to be equivalent to gradient vanishing, i.e., that gradients vanish whenever the proposed similarity measure vanishes and vice versa. In fact, very similar statements to Theorem 1 (i.e., bounds depending on the weight matrix W) can be made for the gradients directly (see Di Giovanni et al, 2024 for instance). Of course it can happen that in the pre-asymptotic case this gradient similarity measure is small or even zero at some point, but this does not denote an issue.

This has to be addressed, otherwise this paper cannot be published. In particular, the authors have to show that there exists an asymptotic regime, where the similarity measure in Theorem 1 goes to zero, but the gradient norms do not. I suspect there are no such cases, but I am happy to be convinced otherwise. The same has to be done for Theorem 2.

That being said, this has also be demonstrated empirically. The provided plots in the paper only show the effect of varying the depth on the similarity measure. The same has to be done for the gradient norms for the exact same setup, i.e., same architecture with same weights. This would demonstrate that there are in fact cases, where gradients oversmooth but do not vanish.

Other issues:
* Some mathematical statements are wrong. For instance, the sentence after equation (3). Here, if B would be perpendicular to span($\bf 1$), it should be in $\mathbb{R}^{N-1,N}$, not in $\mathbb{R}^{N-1,N-1}$. Moreover, this sentence does not add anything to the context and is simply taken from Wu et al. which used it to demonstrate the similarity between this measure and another oversmoothing measure in the literature.
* most of the theory seems to be taken from other papers. It would be good if the authors could state explicitly what is their own theoretical contribution
* it would be good to extend to nonlinear layers, since in practice linear GNNs are not very common
* Only four small-scale graph datasets are considered. It would strengthen the paper if this would be extended.

Minor issues:
* The writing needs to be revised. There are many typos.

**Questions:**

Can you try some of the experiments with GNN architectures that are known to not suffer from gradient vanishing/explosion or oversmoothing? If your gradient similarity measure would go to zero or explode for these cases, it would strengthen your claim.

---

### Author Response · Authors · 2024-11-22
**Withdrawal of Submission and Thanks to Reviewers**

We have decided to withdraw our submission to further develop the work based on the valuable insights gained from the reviews. We sincerely appreciate all the reviewers for their constructive feedback and time.

---

### Note · Authors · 2024-11-22

**Comment:**

We have decided to withdraw our submission to further develop the work based on the valuable insights gained from the reviews. We sincerely appreciate all the reviewers for their constructive feedback and time.

**Withdrawal Confirmation:**

I have read and agree with the venue's withdrawal policy on behalf of myself and my co-authors.